# Numerical simulation study on optimization of key technical parameters of unpowered dust removal system in a gas-solid two-phase flow field

Yuxuan Liu[1], Chen Lv[1,2]*, Sheng Xue[2]

**1** College of Energy Environment and Safety Engineering, China Jiliang University, Hangzhou, China,
**2** Key Laboratory of Safety and High-efficiency Coal Mining, the Ministry of Education (Anhui University of Science and Technology), Huainan, China

* lvchen0707@cjlu.edu.cn

## Abstract

Based on the engineering background of the Huaibei Coal Preparation Plant in Anhui Province, China, this study aims to effectively mitigate dust pollution during the blanking process at coal transfer points. Given the limitations of conventional dust control measures, a numerical simulation was conducted using the Euler model and discrete element method (DEM) for simulating particulate matter. The simulation incorporated the two-way coupling effect between gas-solid two-phase flow and the collision-adhesion dynamics among particles. We examined the migration behavior of dust within an unpowered dust removal system under various technical parameters. Results indicate that installing a return pipe significantly reduces dust overflow caused by impact airflow during blanking. Specifically, when the return pipe diameter is 500 mm, and the horizontal distance between the drainage port of the return pipe and the bottom of the blanking pipe is 2000 mm, the dust removal efficiency reaches its optimal level. Field tests confirmed that after implementing the improved dust removal system, the ambient air dust concentration decreased to less than 2 mg/m³, representing a reduction of approximately 92.02% compared to pre-transformation levels. This approach overcomes the limitations of traditional dust prevention technologies, inhibits dust generation at its source within the coal conveying system, effectively reduces working space dust concentrations, and ensures occupational health and safety for workers.

## 0 Introduction

In the daily production process, the generation of various types of dust is inevitable. When the aerodynamic diameter of harmful dust particles is below 7.07 μm, these fine particulates can enter the human respiratory system through inhalation, leading to pneumoconiosis in workers over time. According to relevant statistics, there

**Data availability statement:** All relevant data are within the paper.

**Funding:** This research was supported by National College Student Innovation and Entrepreneurship Training Program(Grant No.2100603122), Natural Science Foundation of Zhejiang Province (Grant No. LQ20E040005), the State Key Laboratories Program of China (Grant No. JYBSYS2019102).

**Competing interests:** The authors have declared that no competing interests exist.

were over 975,000 cases of pneumoconiosis in 2019, including 873,000 cases of occupational pneumoconiosis, accounting for approximately 90% of all occupational diseases in China. Consequently, pneumoconiosis caused by excessive dust concentration has become the most severe occupational hazard in the country. In industries such as mining, power plants, and material terminals, belt conveyors are widely used due to their advantages of high capacity, efficient material transmission, and ease of installation and operation. However, during the material transfer process on belt conveyors, significant amounts of dust are generated when materials fall at each transfer point. Dust concentration measurements at operational sites indicate that dust levels at conveyor transfer points typically exceed safety standards, posing substantial risks to enterprise safety and employee health. This issue represents a critical technical challenge that production enterprises must urgently address.

Since the 1960s, numerous domestic and international scholars have initiated research on dust control technology. As awareness of the hazards associated with dust has increased, significant advancements have been made in this field. Various innovative dust removal equipment, technologies, and materials have been continuously introduced [1–4]. In 1986, American Martin Engineering Company [5] developed Inertial Flow Technology (IFT). This technology focuses on designing a novel curved coal drop system. Controlling the trajectory of falling material to align closely with the inner wall of the coal drop system forms a stable material flow, thereby reducing collisions between the material and the inner wall and among material particles, thus minimizing dust generation. The design of the coal drop system ensures that the material's exit velocity approximates the belt speed, thereby reducing induced wind speed and suppressing dust dispersion. With the advancement of discrete element method (DEM) simulation software and the enhancement of computer computing power, Cleary, Nordell, et al. employed DEM simulations to model material transport trajectories under actual working conditions. Their results demonstrated that designing the coal drop system in the transport system as a curve can more effectively control the flow direction of discrete materials and reduce the final material outlet velocity, thereby minimizing material impact and dust generation [6]. Building on the original cyclone dust collector design, H. J. Luckner [7] introduced additional modifications to enhance its efficiency for power plants. This improvement aimed to address the low dust removal efficiency despite mechanical dust removal technology's simplicity and ease of maintenance. Test results showed that the modified design expanded the range of captured particle sizes and significantly improved the overall dust removal efficiency of the collector. Xiang Xu [8] combined plasma with a ring electrode through capacitive coupling to refine the original electrode design and voltage levels, thereby enhancing dust removal efficiency. Experimental findings indicated that while dust removal efficiency increased, the required dust removal time increased for larger dust particles. Heinz Antes [9] introduced circulating electrostatic dust removal technology into the mine belt transportation system to reduce dust diffusion in large spaces. Experimental results demonstrated that this technology can achieve a dust reduction efficiency of up to 90% in the workplace. However, it also involves high power consumption and significant manual maintenance requirements,

increasing operational costs. Shi Jinhua et al. [10] also applied spray sprinkler technology to control dust in the belt coal conveying system, achieving satisfactory dust suppression effects. Due to climatic limitations, the dust removal equipment and pipelines often freeze during cold winters, causing water conveyance and spray systems to malfunction. Zhang Wenbin et al. [11] proposed using bag filter dust removal methods to reduce dust concentrations to meet relevant national standard. Nevertheless, this approach suffers from the short service life of the cloth bags, necessitating timely replacement after accumulating a certain amount of coal dust, thereby increasing the maintenance workload. Moreover, inadequate or damaged bag sealing can increase coal dust pollution. Some scholars [12] have suggested employing micro-power dust removal technology for dust control. However, this method also incurs high power consumption and requires substantial manual maintenance, contributing to elevated operational costs.

In summary, the current domestic and foreign scholars 'research on dust removal technology mainly focuses on the technical improvement of the original dust removal device and the comprehensive application of the existing dust removal technology. Given the characteristics of the working conditions of the transshipment point of the coal conveying system, aiming at the problem of excessive dust control at the material transshipment point, the author takes Huaibei Coal Preparation Plant in Anhui Province of China as the research object. Based on the dust generation mechanism analysis and migration law, the idea of applying the curved coal drop pipe combined with the unpowered dust removal device to comprehensively control the problem of excessive dust at the transshipment point is proposed.

## 1  Dust production mechanism

Regarding the formation and growth mechanism of respirable dust, can be categorized into three modes, as illustrated in Fig 1 [13]: nuclear mode, accumulation mode, and coarse mode. The nuclear mode primarily consists of particles with an aerodynamic diameter of less than 0.1μm. In the atmosphere, the nuclear mode originates mainly from the combustion and oxidation processes of combustibles, with a minor contribution from mechanical activities. Accumulation mode particles have an aerodynamic diameter ranging of between 0.1μm and 2.5μm. Apart from the aforementioned generation methods, they are also produced by collision, coagulation, and growth of nuclear mode particles. Due to their small size range, gravitational sedimentation has minimal impact on these particles; therefore, they remain suspended in air for long periods of time, posing a significant challenge for air pollution control efforts. Researchers commonly refer to them as 'bimodal' particles; conventional strategies for controlling such 'bimodal' particles involve promoting their continued condensation and growth into 'coarse mode' particles with an aerodynamic diameter greater than 2.5μm for subsequent treatment.

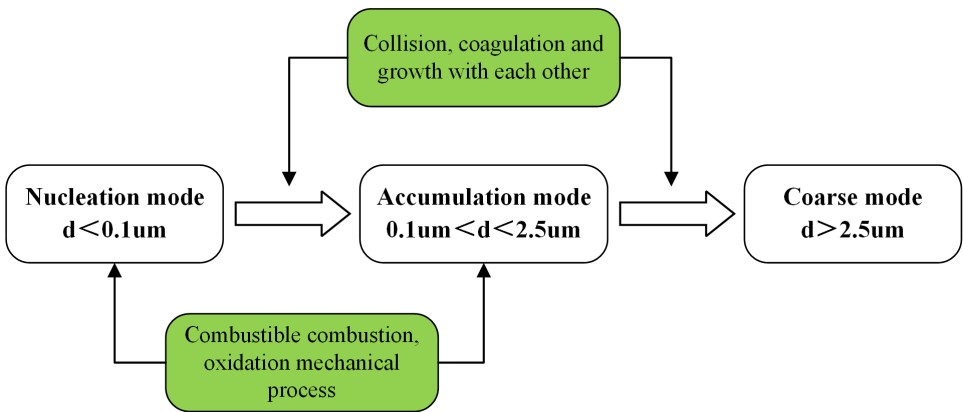

**Fig 1.  The particle size distribution and source of respirable dust.**

However, the natural transformation of 'bimodal' particles into the 'coarse mode' remains a significant challenge. Once in the air, these particles contribute to severe regional environmental pollution and pose a serious threat to human health [14]. To safeguard public health and ecological sustainability, it is crucial to focus on source control of respirable dust. This necessitates the exploration of effective wetting techniques for sub-PM10 dust at its source to promote particle coagulation and growth. In the '14th Five-Year Plan', emphasis is placed on resource conservation as well as the control of fine particulate matter in industries such as coal production. In sectors such as mining, metallurgy, and thermal power generation, the development of efficient methods to capture respirable dust during production is essential for enhancing safety measures and protecting the health of workers. As a result of these efforts, not only the environment is protected, but also the social benefits are maximized.The coal transfer process at the blanking point is accompanied by coal flow dusting, where dust emissions result from the combined effects of various airflows [15]. As the material separates from the top belt and begins free fall, it accelerates, generating a negative pressure that disturbs the surrounding air and creates a high-speed induced airflow. The velocity gradient between this induced airflow and the falling coal causes fine dust particles adhering to the coal surfaces to become airborne under the shear forces exerted by the airflow, resulting in the formation of a high-speed dust-laden flow. As the induced airflow moves downwards through the blanking tube, it compresses, generating a high-pressure environment near the bottom. Due to inadequate sealing in existing guide grooves, significant amounts of dust-laden air escape through gaps into the external environment. When the high-speed material impacts the lower belt, entrapped air between coal particles is released, producing high-speed shear airflow that entrains dust particles. Simultaneously, the collision between the coal flow and the conveyor belt generates high-velocity impact airflows, creating positive pressure vortices within the guide grooves. These high-pressure impact flows and vortices carry dust particles through gaps in the grooves, polluting the working environment. Additionally, as the conveyor belt moves, it drags surrounding air, forming traction flows that further contribute to dust generation. To address the severe dust pollution at transfer points and to improve upon the inefficiency of conventional dust control methods, this study proposes a novel design for non-powered dust removal devices.

## 2  Introduction of dust removal principle of unpowered dust removal system

This product is designed in the field of industrial ventilation and dust removal. By flexibly applying aerodynamic principles, as well as the theories of pressure balance and resistance balance in the ventilation pipe network, the airflow containing dust is effectively organized and adjusted. It skillfully introduces the dust-laden airflow into the entrance section of a closed guide trough, forming an efficient pressure and wind speed adjustment system within it. This achieves decompression, speed reduction, and effective dust reduction, resulting in significant improvements in occupational health and safety for operators (Fig 2). Additionally, to address issues such as harsh operating conditions, high operating costs, and environmental damage caused by traditional dust reduction measures with poor effectiveness, we propose a novel design concept for a reflux-compensated micro-dynamic dust removal device that combines functional modules including closed guide grooves, tail relief chambers, centering devices, built-in damping beds, reflux micro-dynamic compensators, and pressure relief spoiler S-type dampers.

## 3  Geometric model establishment and solution parameter setting

### 3.1  Physical model and meshing

The equal-proportion simplified physical model was established using SolidWorks modeling software, as illustrated in Fig 3. Subsequently, this model was imported into Ansys-Fluent 2023R1 numerical simulation software to perform the subsequent simulation calculations.

In the computational grid division of the physical model, an unstructured grid division algorithm is utilized to discretize the domain with varying cell sizes. Cell sizes range from 60 mm to 120 mm in increments of 10 mm. Mesh quality is

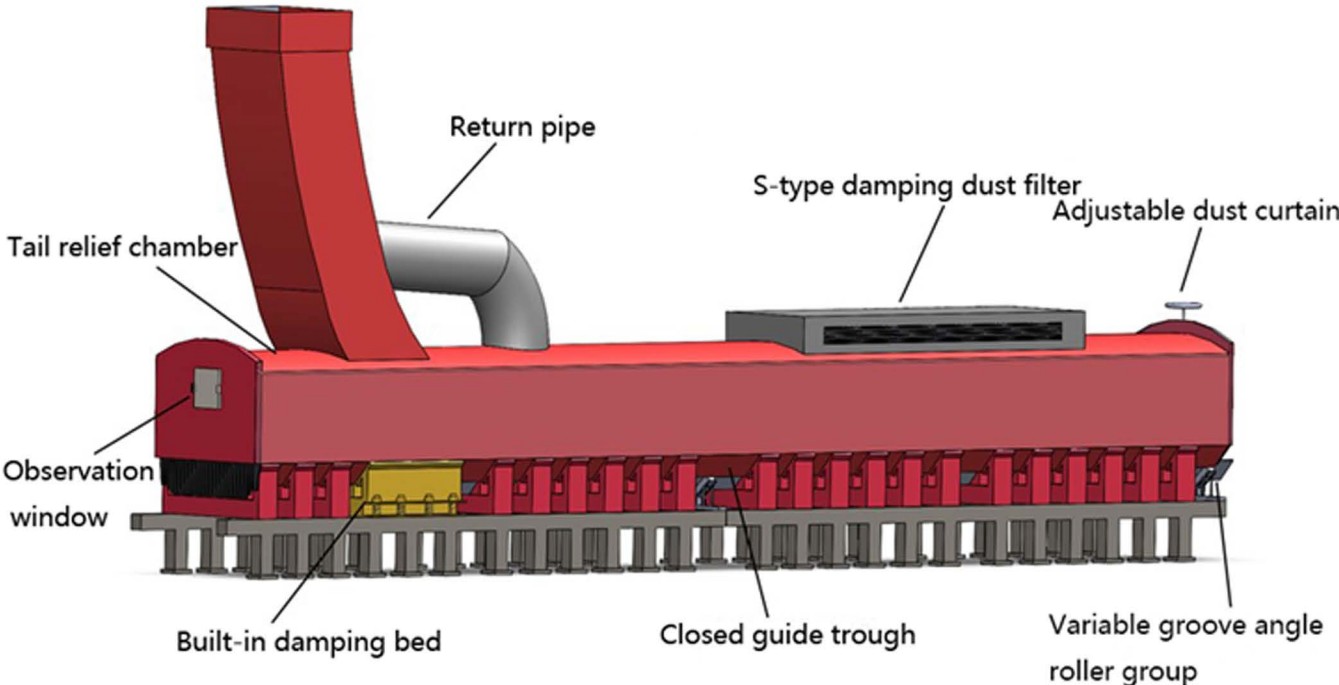

**Fig 2. Unpowered dust removal device stereogram.**

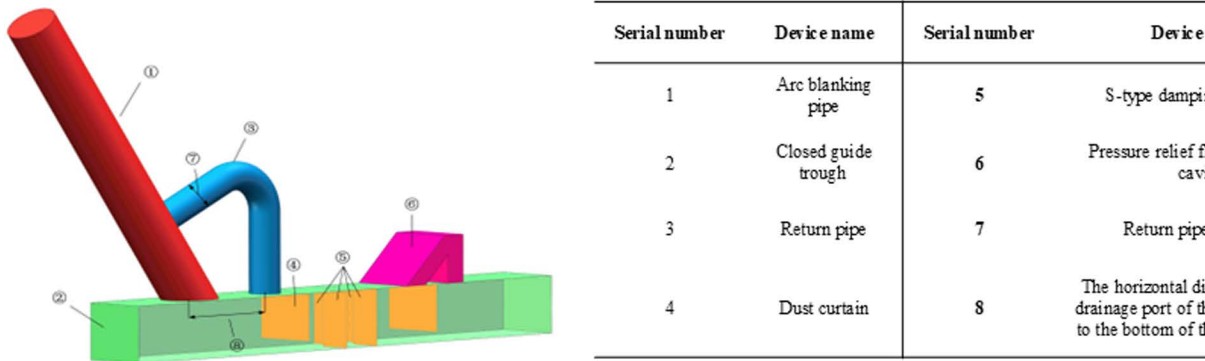

| Serial number | Device name | Serial number | Device name |
|---|---|---|---|
| 1 | Arc blanking pipe | 5 | S-type damping dust filter |
| 2 | Closed guide trough | 6 | Pressure relief flow around the cavity |
| 3 | Return pipe | 7 | Return pipe diameter |
| 4 | Dust curtain | 8 | The horizontal distance from the drainage port of the backflow tube to the bottom of the blanking tube |

**Fig 3. Physical model.**

rigorously evaluated using established criteria such as Skewness, Element Quality, Aspect Ratio, Warping Factor, and Jacobian Ratio, et al. Wind speed and wind pressure monitoring points are strategically placed within the model to examine the impact of different grid sizes on simulation accuracy. This enables the acquisition of pressure and wind speed curves at each monitoring point. The results indicate that when the grid size is less than 100 mm, the convergence of wind speed and wind pressure calculation results from each monitoring point is satisfactory, with deviations in monitoring data across different grid sizes controlled within 5%. Therefore, considering both the accuracy of the calculation results and the efficient use of computing resources, this study ultimately adopted a grid size of 100 mm as the standard. The final grid division outcome is presented in Fig 4.

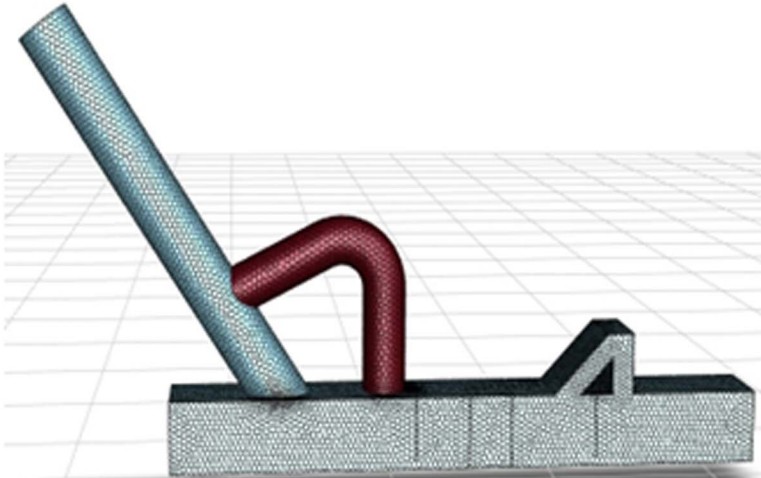

**Fig 4. Grid division diagram.**

## 3.2 Parameter setting

According to the fundamental principle of turbulence in numerical simulation, and considering the specific circumstances of coal transfer points, the model parameters were established as presented in Table 1. Among them, considering the complex flow characteristics of dust-laden airflow in the dust removal pipeline, which may lead to issues such as separation and particle collisions, the Non-Equilibrium Wall Functions method is employed to model accurately and process wall turbulence.

According to the field sampling test, the particle size distribution of dust at the operational site is presented in Table 2, while the discrete phase parameters for numerical simulation are set as illustrated in Table 3 below.

## 4 Gas-solid two-phase flow theory calculation equation

The blanking process at the coal transfer point is characterized as a gas-solid two-phase flow, with air acting as the continuous phase. Given that the volume fraction of coal particles does not exceed one-third of the total flow field volume, the Discrete Phase Model (DPM) is deemed suitable for computation. The airflow field power generated during the unloading process at the transfer point primarily stems from the induced airflow caused by coal particle movement. Therefore, the

**Table 1. Model parameter setting.**

| Parameter | Parameter setting | Parameter | Parameter setting |
|---|---|---|---|
| Solver | Pressure basis solver | Entrance and exit pressure | 0 Pa |
| Turbulent flow model | Standard k-ε model | Turbulent intensity | 4% |
| Entrance boundary type | Pressure inlet | Solution method | SIMPLEC |
| Exit boundary type | Pressure outlet | Convergence criteria | 1e-06 |
| Wall function | Non-Equilibrium Wall Functions | Time step size | 0.001 s |

**Table 2. Dust particle size distribution.**

| Particle size | <2μm | 2～5μm | 5～10μm | 10～20μm | >20μm |
|---|---|---|---|---|---|
| Dispersion degree | 28% | 33% | 24% | 12% | 3% |

 

**Table 3. Discrete phase parameter setting.**

| Discrete phase parameters of coal block | Parameter setting | Dust discrete phase parameters | Parameter setting |
|---|---|---|---|
| Jet type | Surface | Jet type | Surface |
| Mass flowrate | 400 kg/s | Mass flowrate | 0.0002 kg/s |
| Mean diameter | 30mm | Minimum particle size | $2e^{-6}$m |
| Start time | 0s | Maximum particle size | $100e^{-6}$m |
| Ending time | 10s | Mean particle size | $5e^{-6}$m |

coupling effect between the gas and solid phases must be carefully considered in the simulation. Additionally, during the blanking process, collisions between coal particles can significantly influence the coal particles' motion and the flow field's distribution. Consequently, it is crucial to incorporate the particle collision model into the analysis. This study investigates the phenomenon by activating the DEM Collision Model option during the physical model setup. Since the airflow field within the dust collector reaches a steady state through a transient process, the model is solved using an unsteady-state approach. Considering the high precision and broad applicability of the standard k-ε turbulence model [16] and the specific characteristics of the blanking process, the standard k-ε turbulence model is selected for the simulation.

### 4.1 Continuous phase analysis

The Euler method is employed in this study to represent the motion parameters of the airflow field in the dust removal device. Given that heat transfer is not involved in the flow process, the fundamental physical principles encompass mass conservation and momentum laws. Neglecting variations in the density of the dusty airflow allows simplification to an incompressible viscous fluid motion problem. The mathematical representation of this control equation is as follows:

The continuity equation [17]:

$$\frac{\partial \rho}{\partial t} + \frac{\partial}{\partial x_i}(\rho \mu_i) = 0$$

(1)

The momentum equation [18]:

$$\frac{\partial}{\partial t}(\rho \mu_i) + \frac{\partial}{\partial x_j}(\rho \mu_i \mu_j) = -\frac{\partial p}{\partial x_i} + \frac{\partial \tau_{ij}}{\partial x_j} + \rho g_i - f_i$$

(2)

In the formula, $\rho$ represents the fluid density, $x_i$ and $x_j$ denote the coordinates in the three-dimensional space ($x$, $y$ and $z$), $\mu_i$ and $\mu_j$ represent the velocity components along each coordinate direction ($x$, $y$ and $z$), $p$ signifies the fluid pressure, $\tau_{ij}$ denotes the stress tensor, $f_i$ represents the resistance of the fluid in each coordinate direction ($x$, $y$ and $z$), while $g$ stands for gravitational acceleration.

### 4.2 Discrete phase analysis

Dust is subject to multiple forces during its migration and diffusion in the airflow field. The equation of motion for a single dust particle can be expressed as follows:

$$\sum F = m_p \frac{d\nu_p}{dt}$$

(3)

In the formula, $\sum F$ represents the cumulative external force acting on an individual dust particle, where $m_p$ denotes the mass of the particle and $v_p$ signifies its velocity. The impact of forces on dust migration and diffusion in the flow field

varies. Among these forces, pressure gradient force, additional mass force, and Basset force are considered negligible; however, gravity, buoyancy, aerodynamic drag, Saffman lift, and Magnus force significantly influence dust movement within a dust collector's flow field. Accurate characterization of particle behavior necessitates their precise calculation. During the migration of dust particles, collisions occur between particles and between particles and walls. The calculation equation for soft elastic collisions can be derived based on the methodology presented in Reference [19].

### 4.3 Characterization parameters of dust removal effect

From the perspective of dust, in conjunction with the numerical simulation results obtained through the Discrete Phase Model (DPM), the characterization parameter for evaluating the efficiency of dust removal by the dust collector is defined as the rate of dust deposition. The corresponding theoretical equation can be expressed as follows.

$$\eta = \frac{N_s}{N_t}$$

(4)

In the equation, $\eta$ represents the rate of dust deposition, $N_t$ denotes the total number of dust particles generated within the enclosed guide trough, and $N_s$ signifies the quantity of deposited dust particles.

Focusing on the fluid in the reflux pipe as our research subject, we can derive from mass flow conservation principles:

$$\rho \nu_1 S_1 = \rho v_2 S_2$$

(5)

In the equation, $\rho$ represents the air density, $\nu_1$, and $\nu_2$ denote the fluid velocity at both ends of the return pipe cross-section, while $S_1$ and $S_2$ indicate the respective areas of these cross-sections.

Because the air density is constant, the cross-sectional area at both ends of the reflux pipe is $S_1 \approx S_2$, so $\nu_1 \approx \nu_2$. According to the Bernoulli equation:

$$p_1 + \frac{1}{2}\rho \nu_1^2 + \rho g h_1 = p_2 + \frac{1}{2}\rho \nu_2^2 + \rho g h_2 + \Delta E_f$$

(6)

The formula is expressed as follows: $p_1$ and $p_2$ represent the static pressure at both ends of the reflux pipe, $h_1$ and $h_2$ denote the height of the cross section at both ends, while $\Delta E_f$ represents the energy loss per unit volume.

Formula (6) can be simplified as:

$$\Delta E_f = (p_1 - p_2) + \frac{1}{2}\rho \left( \nu_1^2 - \nu_2^2 \right) + \rho g(h_2 - h_1)$$

(7)

In the formula, due to the approximate equality of $\nu_1$ and $\nu_2$, the change in fluid kinetic energy can be disregarded. Additionally, the cross-sectional height remains constant at both ends of the return pipe, resulting in a constant change in fluid gravitational potential energy. From this analysis, it is evident that the energy loss per unit volume $\Delta E_f$ is positively correlated with the pressure difference per unit volume $p_1 - p_2$. The energy loss $\Delta E_f$ at both ends of the return pipe signifies the impact of pressure reduction and deceleration on dust collection after high-pressure and high-speed dust-laden airflow enters into it. Therefore, a larger value for $p_1 - p_2$ amplifies the effect of pressure reduction and deceleration on dust collection while increasing its likelihood to settle.

## 5 The influence of key technical parameters on the dust removal efficiency of dust collector

We measured the wind speed and pressure distribution at different cross-section positions along the airflow direction. By analyzing the stability of the velocity profile along the wind speed flow direction and the pressure drop characteristics

along the flow direction, we can determine whether the internal flow of the dust collector has been fully developed. Based on the full development of the internal flow of the unpowered dust removal system, we carried out a numerical simulation analysis on the relevant physical parameters affecting the dust removal efficiency. The specific research results are as follows.

### 5.1 Analysis of the influence of return pipe on dust removal efficiency of dust collector

In order to investigate the impact of installing the return pipe assembly on the dust removal efficiency of the dust collector, a numerical simulation is conducted to analyze the airflow field in the closed guide tank without the return pipe. The simulation results provide velocity and pressure contours of the airflow under this operating condition, as illustrated in Figs 5 and 6.

From the analysis of the velocity cloud diagram as shown in Fig 5, it is evident that during the accelerated falling process of the coal flow, the disturbance in the nearby flow field generates a high-speed induced airflow within the blanking tube. When the material flow strikes the bottom belt, a high-speed impact airflow is produced, and at the bottom of the guide groove, high-speed swirls are generated due to wall blockage. At this point, the wind speed at the outlet of the guide groove can reach 1.05 m/s. According to the pressure cloud diagram analysis as shown in Fig 6, the positive pressure at the junction of the coal conveying belt and the coal drop pipe reaches its peak, leading to a sharp increase in air pressure at the blanking point. Due to the limited release space, the air volume induced by the primary dusting effect also reaches its maximum. Upon collision with the belt, the coal flow generates a strong air shock wave, causing the dust-laden air mass to spread evenly around the blanking pipe, resulting in severe dust dispersion. This phenomenon makes it easy for dust and powder to escape from this area. Simultaneously, the surrounding air is disturbed again and induced to form a secondary airflow under the pressure exerted by the subsequent coal flow. The secondary airflow combines with other airflows, diffusing the dust-laden air mass into the surrounding environment. The dust source fills the entire working space through repeated, continuous action.

To control dust generation at its source, a non-powered dust removal technology is proposed to transform transfer points in existing belt coal conveying systems based on coal dust's generation mechanism and diffusion characteristics.

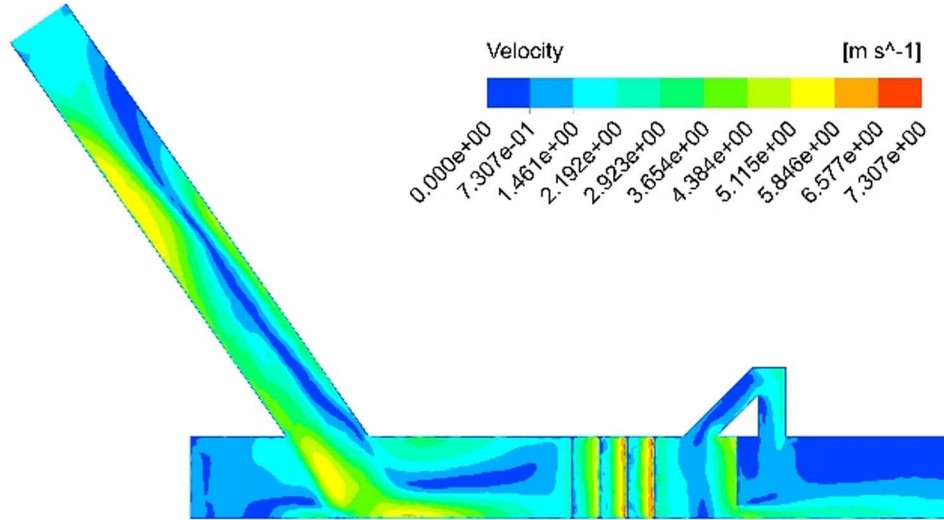

**Fig 5. Simulation results of the velocity contour for the no-return pipe.**

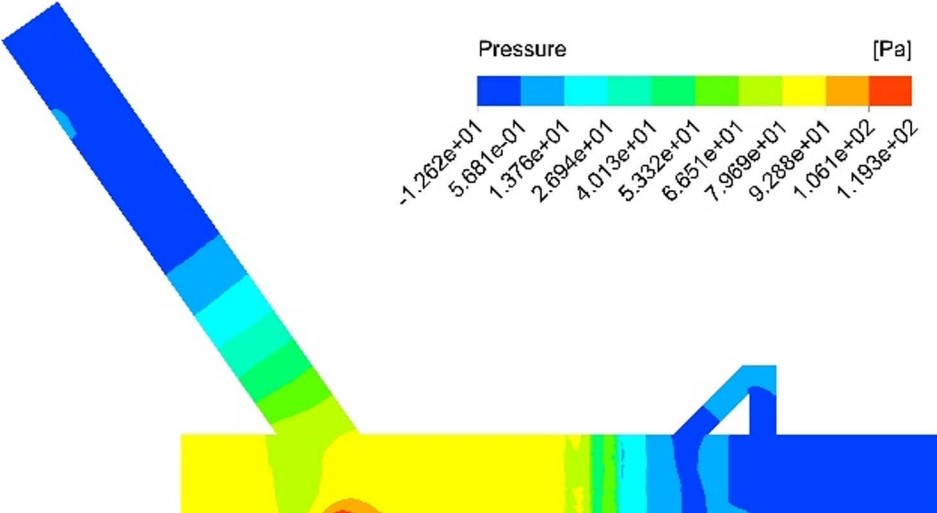

**Fig 6. Simulation results of the pressure contour for the no-return pipe.**

This technology primarily employs aerodynamics and pressure equilibrium principles, utilizing a closed-loop circulation balance device (reflux pipe) to control material flow. By balancing the induced wind pressure inside the guide groove with the external space pressure, this method effectively mitigates the high-pressure environment and high-speed induced airflow and impact airflow generated during the blanking process, thus preventing dust spillage caused by poor sealing of the guide groove. The key advantage of this technology lies in its ability to significantly reduce the velocity of the gas-solid two-phase medium without any external power, thereby inhibiting dust generation during the material falling process. Numerical simulations have been conducted to obtain the velocity and pressure contours of the airflow field under this working condition, as illustrated in Figs 7 and 8.

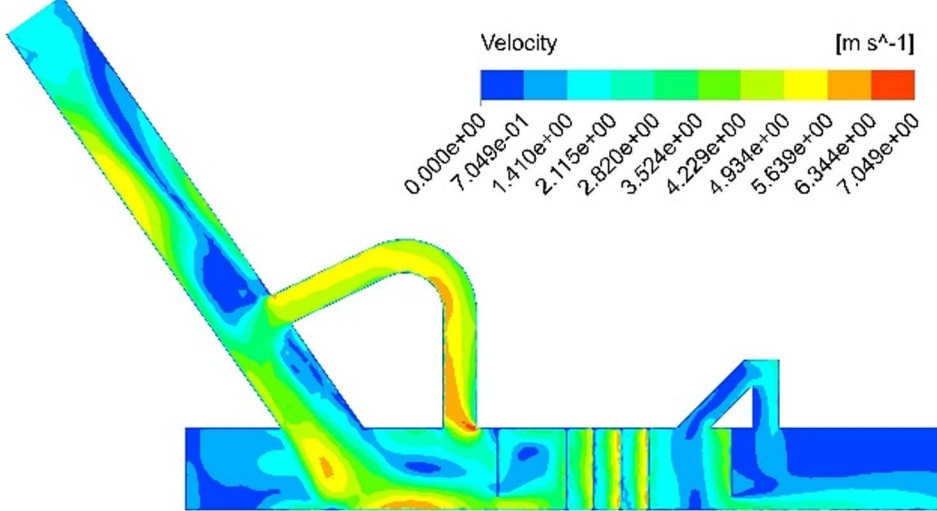

**Fig 7. Simulation results of the velocity contour following the installation of return pipe.**

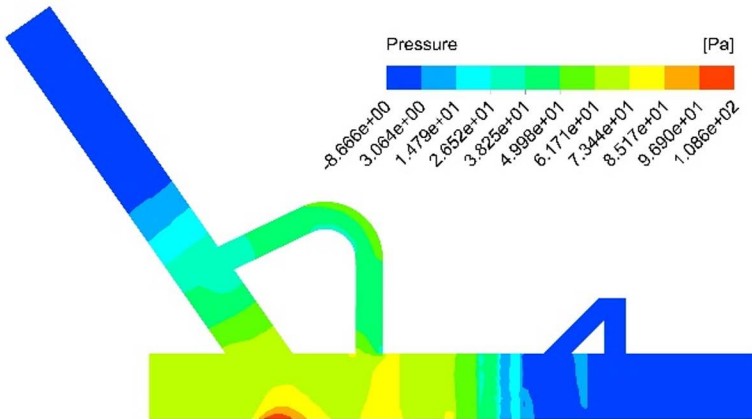

**Fig 8. Simulation results of the pressure contour following the installation of return pipe.**

The velocity cloud diagram as shown in Fig 7 reveals that the installation of the return pipe leads to a reduction in wind speed at the tail outlet of the closed guide groove, reaching 0.79 m/s. Additionally, with guidance from the drainage plate, impact airflow at the bottom of the blanking pipe enters the return pipe. This flow disturbance and neutralization around the return pipe effectively mitigates the promoting effect of induced airflow on high-pressure conditions at the bottom of the guide groove. Furthermore, when considering pressure distribution through a pressure cloud diagram as shown in Fig 8, it is evident that there is a significant decrease in pressure within the flow field at the bottom of the guide groove. Consequently, this approach can effectively address dust spillover issues caused by positive high-pressure environments at its base.

The discrete element method (DEM) model is utilized to simulate the movement of coal flow particles, with the trajectory of material particles illustrated in Fig 9. As shown in the diagram, after dust enters the reflux tube under the influence of the drainage plate, its kinetic energy decreases due to particle collisions. Following collisions with the tube wall, the particles adhere to the surface and gradually coalesce into larger particle clusters. Guided by airflow and gravity, these particles settle naturally. By comparing dust deposition and escape under two working conditions—with and without the return pipe (as shown in Table 4)—the results demonstrate that the return pipe's installation significantly enhances the device's dust removal efficiency by 24.3%.

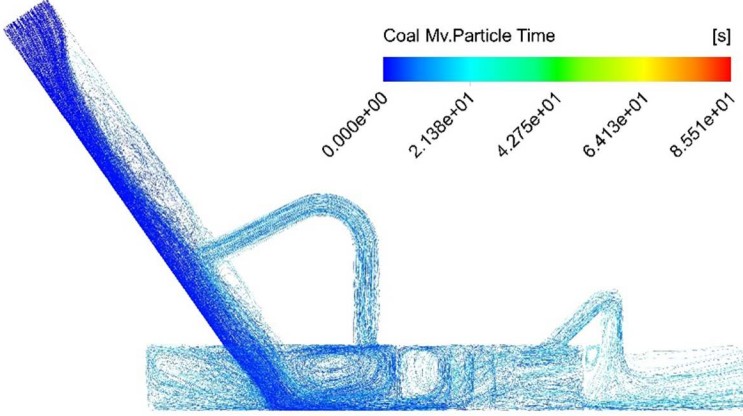

**Fig 9. Dust particle flow trace diagram.**

**Table 4. Dust deposition, escape data table.**

| Module | Total number of particles | Settlement number | Escape number | Particle collection efficiency/ % |
|---|---|---|---|---|
| No return tube | 10799 | 6101 | 4698 | 56.5 |
| Return tube | 10786 | 8715 | 2071 | 80.8 |

## 5.2 Analysis of the influence of the diameter of the return pipe on the dust removal effect of the dust collector

In order to study the influence of the diameter of the return pipe on the dust removal effect of the dust collector, five groups of numerical simulations were set up with the diameter of the return pipe of 400 mm, 500 mm, 600 mm, 700 mm and 800 mm respectively. Through simulation, the curves of the pressure difference between the two ends of the return pipe of the unpowered dust collector, the maximum wind speed in the closed guide groove and the dust removal efficiency with the diameter of the return pipe are obtained, and the pressure difference under different diameters is sorted from small to large, and the corresponding average dust deposition time is obtained. The curve of the average deposition time with the pressure difference is shown in Figs 10 and 11.

The pressure difference at both ends of the reflux pipe, the maximum wind speed in the closed guide groove, and the dust removal efficiency of the dust collector all exhibit their peak values when the diameter is 500 mm according to Fig 10. Specifically, a maximum pressure difference between the two ends of the return pipe is observed, accompanied by a minimum value for the maximum wind speed in the closed guide tank and a maximum value for dust removal efficiency of the dust collector. Moreover, as the pressure difference increases, there is a gradual decrease in the average deposition time of dust particles with its shortest duration occurring at maximal pressure difference.

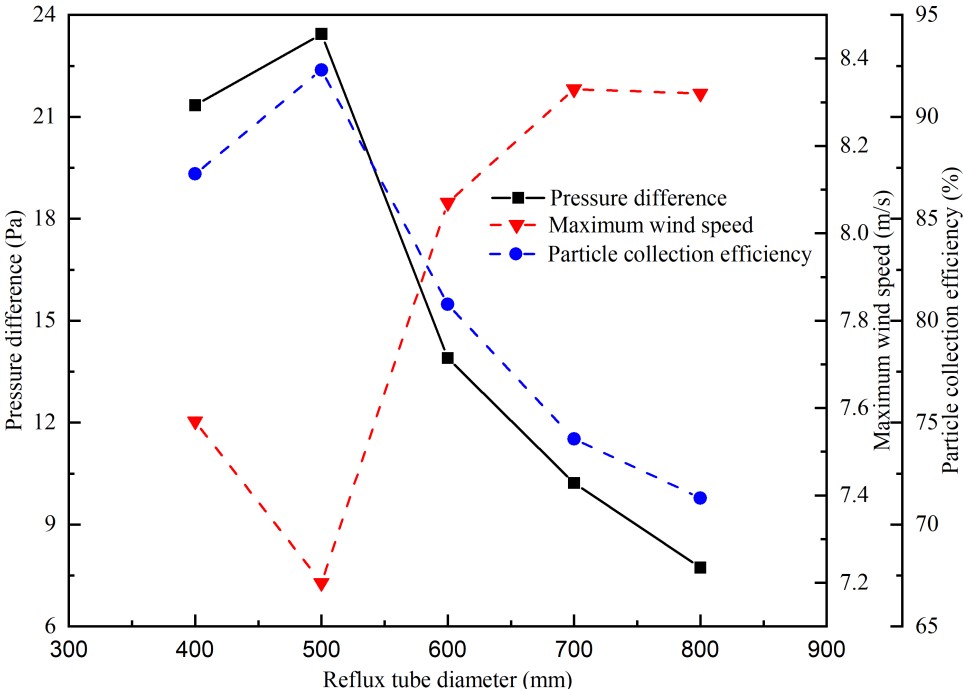

**Fig 10. The relationship between pressure difference, maximum wind speed, dust removal efficiency.**

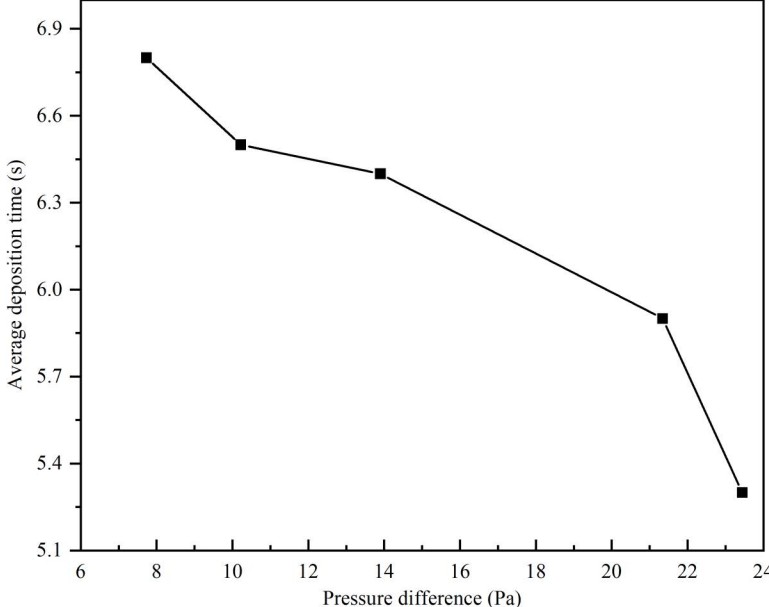

**Fig 11. The relationship between average deposition time and pressure difference.**

Based on the characterization parameters of the dust removal efficiency of the dust collector in Fig 11, an increased pressure difference across the return pipe leads to more significant fluid energy loss. When the return pipe diameter is 500 mm, the pressure difference peaks, resulting in a pronounced decompression and deceleration effect within the dust collector. As the velocity of the dust decreases during internal migration, it becomes more likely for the particles to settle or coalesce into larger aggregates, eventually falling onto the surface of the conveyor belt.

## 5.3 Analysis of the influence of the horizontal installation position of the return pipe on the dust removal effect of the precipitator

To investigate the impact of the horizontal installation position of the return pipe on the dust removal efficiency of the dust collector, five sets of numerical simulations were conducted based on a 500 mm diameter return pipe. The horizontal distance from the drainage port of the return pipe to the bottom of the blanking pipe was set at 1400 mm, 1700 mm, 2000 mm, 2300 mm, and 2600 mm respectively. The simulations provided data on pressure differences between the two ends of the unpowered dust collector's return pipe, the maximum wind speed in the enclosed guide groove, and changes in dust removal efficiency with varying horizontal installation positions for the return pipe. The pressure differences at different horizontal distances were sorted from smallest to largest and corresponding average deposition times were determined. Figs 12 and 13 show a graph depicting the average deposition time against the pressure difference.

As shown in Fig 12, it is evident that the pressure difference at both ends of the reflux pipe, maximum wind speed in the closed guide groove, and dust removal efficiency of the collector all reach their extremes when the horizontal distance is 2000 mm. At this point, there is a significant increase in pressure difference at both ends of the return pipe resulting in improved dust collection efficiency due to a better speed reduction effect. The average deposition time of dust gradually decreases with increasing pressure difference and reaches its minimum value when the pressure difference is highest.

Based on the characterization parameters of the dust removal effect in Fig 13, when the horizontal distance is 2000 mm, the pressure difference across the return pipe reaches its maximum, resulting in the highest fluid energy loss and the most significant pressure and velocity reduction by the dust collector. At this distance, the material can descend

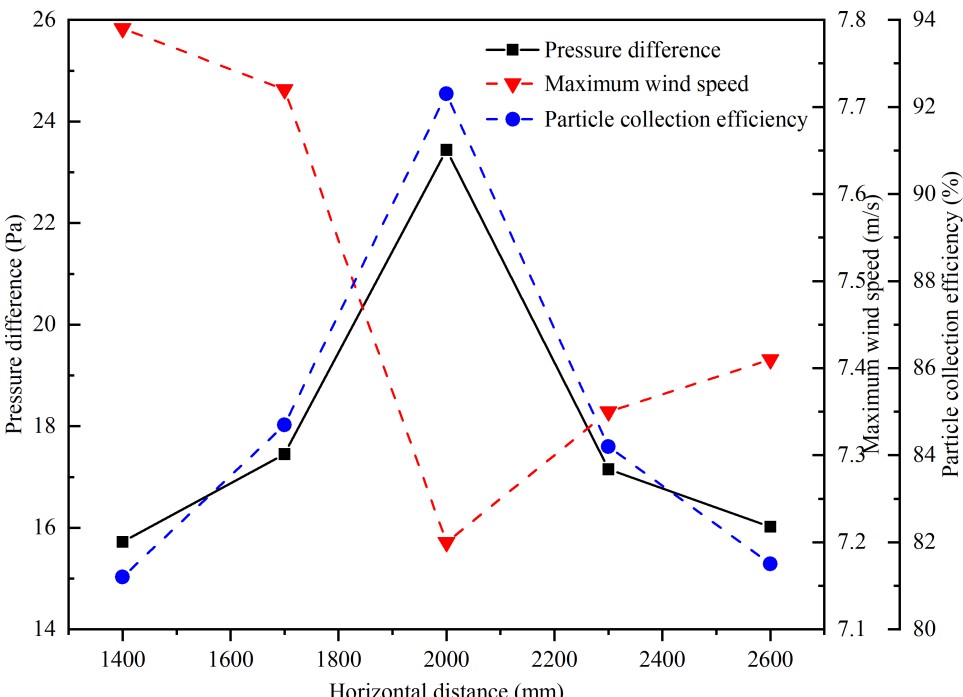

**Fig 12. The relationship between pressure difference, maximum wind speed, dust removal efficiency and horizontal distance.**

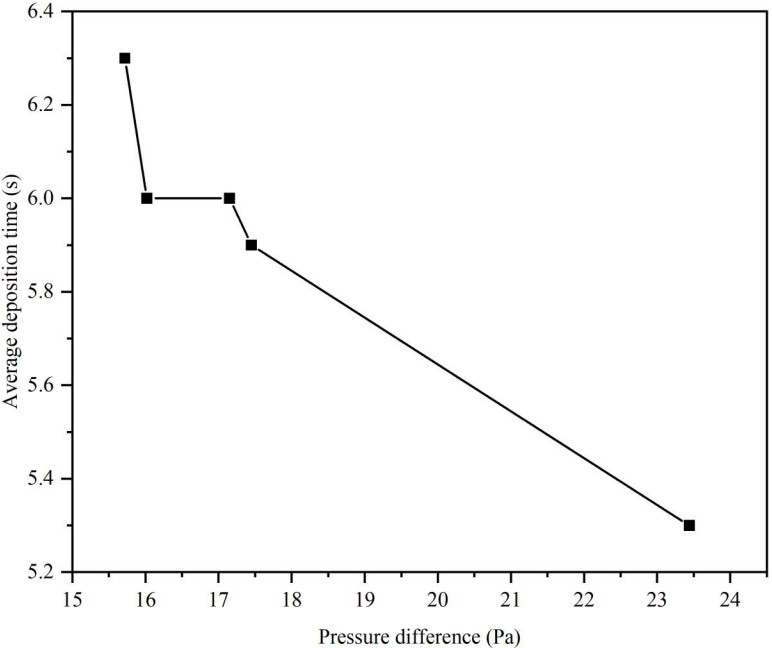

**Fig 13. The relationship between average deposition time and horizontal distance.**

along the coal pipe wall at a regulated speed to minimize dust generation and reduce airflow velocity. This approach, combined with dust deposition treatment at the blanking point, overcomes the limitations of traditional dust removal methods. The dust control mechanism is primarily reflected in the following aspects: the negative pressure airflow generated during the material's descent is channeled through the return pipe, causing the induced airflow to linger within the inertial coal drop pipe and increasing the collision frequency of coal dust particles. Consequently, the kinetic energy of the coal dust particles is converted into potential energy, reducing their velocity and allowing them to gradually settle onto the coal conveyor belt, thereby confining the dust production process to the flow within the coal drop pipe.

## 6 Field application effect analysis

### 6.1 Company profile

The coal preparation plant in Huaibei City initially had a processing capacity of 3.00 million tons per annum (Mt/a). Through continuous upgrades and expansions, this capacity has now increased to 16.00 Mt/a. The plant processes a variety of coal types, including coking coal, fat coal, and 1/3 coking lean coal. The raw coal is sourced from mines such as Linhuan, Haizi, Tongting, Yangliu, Suntuan, Yuandian No.1 well, and Qingdong. In 2017, the total raw coal output from these mines was approximately 18.30 Mt/a, which rose to 20.70 Mt/a in 2018 and further increased to 26.00 Mt/a in 2019, ensuring a reliable supply for ongoing plant expansions. The majority of clean coal products are supplied to leading domestic metallurgical enterprises, including Baosteel and Masteel, as well as major coking plants. By-products, such as coal slime, middling coal, and gangue, are utilized as fuel for power generation and transported to power plants. Additionally, gangue is sent to brick factories to be used as a raw material for brick production, thus contributing to a circular economy that extends the coal industry chain while supporting regional economic development.

### 6.2 Dust sampling test results

The coal handling system's belt conveyor at the plant has been in a state of disrepair for an extended period, leading to misalignment with current production capacity. Furthermore, the inherent limitations of specific specialized mechanical equipment have rendered the existing dust removal system incapable of meeting the required production environmental standards. It is proposed that a transformation be undertaken based on the operational principles of unpowered dust removal technology and parameter optimization simulation results. Through comprehensive on-site investigations and analyses of multiple belt conveyor transfer points, the belt coal conveying systems in typical scenarios, such as the raw coal unit, washing unit, and transfer station within the coal preparation plant's conveying unit, were selected for pilot transformation. Sampling and testing were conducted at each location. By comparing and analyzing the test results, the practical application effects of the unpowered dust removal system utilizing reflux pipes were evaluated (the physical device on site is shown in Fig 14).

The AKFC-92 A mine dust sampler was utilized to employ the filter membrane quality method for detecting and analyzing the contact concentration of respirable coal dust in workplace air. As shown in Table 5, long-term respirable dust concentrations ($C_{TWA}$) for belt drivers before and after the transformation of the transfer station are presented.

### 6.3 Dust control effect analysis

According to the results of dust detection, prior to the systematic transformation of each transfer station in the coal preparation plant, only certain areas of each transfer point met the dust concentration standard. However, in other areas, the dust concentration index exceeded the occupational exposure limit for workplace dust factors, posing a significant threat to workers' occupational health and safety. The underlying cause lies in an unreasonable design of the dust removal system and uncontrolled organization of high-speed and high-pressure airflow at transfer points, which hinders the effective settling of dust particles and consequently leads to severe airborne particulate matter dispersion.

 

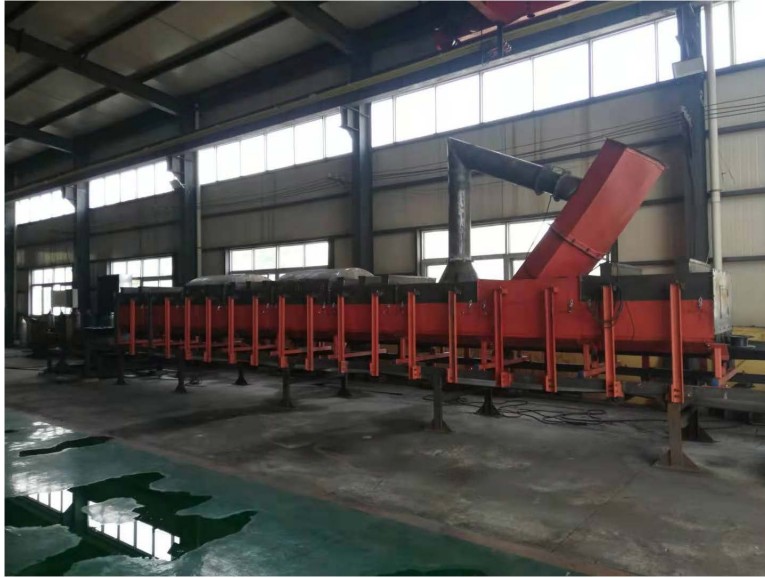

**Fig 14. Unpowered dust removal system field physical map.**

Table 5. Individual long-term respiratory dust concentration (C$_{TWA}$) test results.

| Testing element | Job site | Detected results C$_{TWA}$(mg/m³) | | China standard (mg/m³) | European standard (mg/m³) | American standard (mg/m³) |
|---|---|---|---|---|---|---|
| | | Before transformation | After transformation | | | |
| Raw coal unit | East coal pit | 3.48 | 0.31 | 4 | 3 | 1.8 |
| | Eastern raw coal blending warehouse | **5.52** | 0.44 | 4 | 3 | 1.8 |
| | Western coal pit | **5.12** | 0.39 | 4 | 3 | 1.8 |
| | Western raw coal blending warehouse | 3.56 | 0.32 | 4 | 3 | 1.8 |
| Coal Washing unit | Under East washing building raw coal bunker | **6.6** | 0.52 | 4 | 3 | 1.8 |
| | East washing building roof belt | 3.4 | 0.30 | 4 | 3 | 1.8 |
| | Eastern clean coal bunker allocation warehouse | 3.88 | 0.36 | 4 | 3 | 1.8 |
| | Western District raw coal bunker | **5.08** | 0.76 | 4 | 3 | 1.8 |
| | On the western clean coal bunker | **6.42** | 1.07 | 4 | 3 | 1.8 |
| Coal conveying unit | West coal slime belt | 4.75 | 0.25 | 4 | 3 | 1.8 |

After implementing systematic transformations at each transfer station, the dust concentration at every transfer point was successfully reduced below the occupational exposure limit for dust factors in the workplace. Notably, the long-term respirable dust concentration in the belt driver decreased to 0.25 mg/m$^{-3}$, representing only 5% of the occupational exposure limit. Furthermore, test results from other sampling points also demonstrated compliance with national occupational health standards, confirming that the non-powered dust removal system installed in the return pipe effectively mitigated dust emissions at coal transfer points.

In summary, transforming the unpowered dust removal system in the coal preparation plant has substantially mitigated the issue of dust levels exceeding standards at each transfer point. Post-transformation, the dust concentration at all transshipment points was reduced to below the occupational exposure limits for workplace dust, fully complying with

national and international regulations. This not only effectively safeguards the health and safety of workers but also significantly reduces the risk of occupational pneumoconiosis.

## 7 Conclusions

(1) In this study, we conducted numerical simulations to investigate the airflow dynamics in the enclosed guide trough at the coal transfer point. Our findings reveal that during the material descent from the discharge tube to the lower conveyor belt, a high-speed and high-pressure flow field is generated at the bottom of the guide trough. However, due to inadequate sealing of the guide trough, dust easily overflows and escapes.

(2) To effectively reduce the pressure and velocity, this study employs passive dust removal technology by incorporating a return pipe device between the blanking pipe and the lower guide trough for numerical simulation. An unpowered dust removal device successfully achieves the objective of reducing pressure and velocity, resulting in a significant improvement in dust removal efficiency.

(3) When the diameter of the reflux pipe is 500 mm and the horizontal distance between the drainage port of the reflux pipe and the bottom of the blanking pipe is 2000mm, a maximum pressure difference occurs between the two ends of the reflux pipe, resulting in increased fluid energy loss. This leads to enhanced decompression and deceleration effects on both the induced airflow and the impact airflow, ultimately improving dust removal efficiency in the dust collector.

(4) This paper focuses on investigating and analyzing the optimum dust removal effect of three key parameters concerning the return pipe: the radius of the return pipe and the horizontal distance from the outlet of the return pipe to the bottom of the blanking pipe. It is important to note, however, that other structural parameters also influence the effectiveness of unpowered dust collectors in suppressing dust. Therefore, further research and analysis are required for these additional parameters.

## Author contributions

**Conceptualization:** Sheng Xue.

**Investigation:** Yuxuan Liu.

**Methodology:** Chen Lv.

**Writing – original draft:** Yuxuan Liu.

**Writing – review & editing:** Chen Lv.

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
