## [Decision Letter · Decision Letter 0]

16 Dec 2024

PONE-D-24-47959Numerical Simulation Study on Optimization of Key Technical Parameters of Unpowered Dust Removal System in a Gas-solid Two-phase Flow FieldPLOS ONE

Dear Dr. Lv,

Thank you for submitting your manuscript to PLOS ONE. After careful consideration, we feel that it has merit but does not fully meet PLOS ONE’s publication criteria as it currently stands. Therefore, we invite you to submit a revised version of the manuscript that addresses the points raised during the review process.

**In addition to the reviewer comments, the authors are suggested to consider points given below:**

**The convergence criterion of 0.001 is high. The simulations need to be performed with lower values such as 0.00001 or 0.000001.****The regulatory framework /industry standards for other regions should also be considered.**

We look forward to receiving your revised manuscript.

Kind regards,

Muhammad Shakaib, PhD

Academic Editor

PLOS ONE

**Journal Requirements:**

This research was supported by National College Student Innovation and Entrepreneurship Training Program(Grant No.2100603122), Natural Science Foundation of Zhejiang Province (Grant No. LQ20E040005), the State Key Laboratories Program of China Grant No. JYBSYS2019102).

This research was supported by National College Student Innovation and Entrepreneurship Training Program(Grant No.2100603122), Natural Science Foundation of Zhejiang Province (Grant No. LQ20E040005), the State Key Laboratories Program of China Grant No. JYBSYS2019102.

This research was supported by National College Student Innovation and Entrepreneurship Training Program(Grant No.2100603122), Natural Science Foundation of Zhejiang Province (Grant No. LQ20E040005), the State Key Laboratories Program of China Grant No. JYBSYS2019102).

Reviewers' comments:

Reviewer's Responses to Questions

**Comments to the Author**

1. Is the manuscript technically sound, and do the data support the conclusions?

Reviewer #1: No

Reviewer #2: Yes

2. Has the statistical analysis been performed appropriately and rigorously? 

Reviewer #1: No

Reviewer #2: Yes

3. Have the authors made all data underlying the findings in their manuscript fully available?

Reviewer #1: No

Reviewer #2: Yes

4. Is the manuscript presented in an intelligible fashion and written in standard English?

Reviewer #1: Yes

Reviewer #2: Yes

5. Review Comments to the Author

**Reviewer #1: ** The title of the manuscript carries the words "numerical simulation study" but the author do not mention any in-house code or commercial software package to conduct the CFD simulations.

The vast majority of the references are cited in the introduction. There are no references to support the numerical simulation methodology.

The numerical results should be validated with respect to a peer-reviewed publication.

**Reviewer #2: ** The description and technical processing are satisfactory. However, details of the results and their subsequent discussion are missing from the manuscript. The following queries should be taken up by the authors:

1- The text requires an overall revision regarding grammar and technical terms, so it should be edited specifically for the language by some native editors.

2- In its present form, the abstract of the work does not sound well. The abstract should contain the following: objectives, methods/analysis, findings, and novelty/improvement.

3-The literature review is superficial. There is only a list of papers without any discussion about them. The gap in knowledge should be clearly presented. Even the motivation of the study should be clearly highlighted. At the end of the introduction, introduce the novelty of your study.

4-The innovation of the paper is not completely clear, kindly mention the novelty of the present analysis.

5-Aside from the aim stated in the title, the research gap and the goals of the research are not specified which leads to the reader missing the significance of the research.

6-Why was the standard k-ε turbulence closure model was used the simulations? Did the authors try different turbulence models?

7-The authors need to clarify how they examined if the flow was fully developed.

8-Can authors justify the inclusion of computational mesh in detail? However, the mesh dependency studies are not very convincing. The authors may consider using one of the more sophisticated methods, such as the Grid Convergence Index (GCI), which is commonly applied and recommended by the Fluids Engineering Division of the American Society of Mechanical Engineers as a reliable method for mesh independence analysis. This methodology is described in P. J. Roache, "Perspective: a method for uniform reporting of grid refinement studies," 1994 and applied in much research

9-Can authors justify the inclusion of boundary conditions in detail?

10- Additional details on the reliability and significance of the validation test used to evaluate the differences between experimental and numerical simulation values should be included.

11- The critical discussion part has to be enlarged. You need strong discussion in your study. It has to be improved to pin-point the actual novelty(s) in this study. The results are only presented, but a proper discussion is missing.

6. PLOS authors have the option to publish the peer review history of their article (what does this mean? ). If published, this will include your full peer review and any attached files.

**Do you want your identity to be public for this peer review?** For information about this choice, including consent withdrawal, please see our Privacy Policy .

Reviewer #1: No

Reviewer #2: No

---

## [Author Response · Author response to Decision Letter 1]

22 Jan 2025

Dear Prof. Muhammad Shakaib,

Thank you very much for your letter and the comments from the referees about our paper submitted to Plos One.

We have checked the manuscript and revised it according to the comments. We submit here the revised manuscript as well as a list of changes.

If you have any question about this paper, please don’t hesitate to let me know.

Sincerely yours,

Lv Chen

Response to Editor

Thanks for your comments on our paper. We have revised our paper according to your comments:

(1) The convergence criterion of 0.001 is high. The simulations need to be performed with lower values such as 0.00001 or 0.000001.

Reply According to the editor’s comment, We have refined the convergence criterion and conducted numerical simulations in accordance with the updated criterion.

(2) The regulatory framework /industry standards for other regions should also be considered.

Reply According to the editor’s comment, we also incorporated dust exposure limit standards from other regions, including both European and American standards.

Response to Reviewer 1:

Thanks for your comments on our paper. We have revised our paper according to your comments:

(1) The title of the manuscript carries the words "numerical simulation study" but the author do not mention any in-house code or commercial software package to conduct the CFD simulations.

Reply According to the reviewer’s comment, We have provided additional clarification in the paper. Specifically, we utilized a commercially available and widely recognized software to conduct numerical simulations of the engineering problems associated with the unpowered dust removal system discussed herein.

(2) The vast majority of the references are cited in the introduction. There are no references to support the numerical simulation methodology. The numerical results should be validated with respect to a peer-reviewed publication.

Reply According to the reviewer’s comment, In the introduction, we have incorporated a discussion on the numerical simulation method and its research advancements. However, existing literature offers limited numerical simulation results for this specific engineering problem, making it challenging to provide adequate data validation support. Consequently, our research group intends to construct a similar experimental simulation platform (as shown in Fig. 1) to verify the numerical simulation results and conduct further analysis and discussion.

Response to Reviewer 2:

(1) The text requires an overall revision regarding grammar and technical terms, so it should be edited specifically for the language by some native editors.

Reply According to the reviewer’s comment, We have revised the grammar and technical terms of the text.

(2) In its present form, the abstract of the work does not sound well. The abstract should contain the following: objectives, methods/analysis, findings, and novelty/improvement.

Reply We appreciate the reviewers' guidance on abstract writing and have revised the abstract under your suggestions.

(3) The literature review is superficial. There is only a list of papers without any discussion about them. The gap in knowledge should be clearly presented. Even the motivation of the study should be clearly highlighted. At the end of the introduction, introduce the novelty of your study.

Reply Following the expert's recommendations, we thoroughly revised the introduction section. When citing others' research findings, we provided a critical evaluation and discussed our perspective on these results, as well as identified common challenges encountered in this field of study. At the end of the introduction, we outlined this research's primary focus and innovative aspects.

(4) The innovation of the paper is not completely clear, kindly mention the novelty of the present analysis.

Reply According to the reviewer’s comment, at the end of the introduction, we point out the innovative work of our research.

(5) Aside from the aim stated in the title, the research gap and the goals of the research are not specified which leads to the reader missing the significance of the research.

Reply We appreciate the valuable suggestions provided by the review expert We address the research gap by discussing the findings of previous studies. In the initial section of the introduction, we elucidate the study's objectives within the context of practical engineering applications. As for the significance of the research, we make a supplementary explanation in the last part of the introduction.

(6) Why was the standard k-ε turbulence closure model was used the simulations? Did the authors try different turbulence models?

Reply In the process of numerical simulation and analysis, we also consider some other turbulence models used in the past for comparative analysis. However, in the process of using different models (such as the k-ω model, LES model, etc. ) for calculation, it is found that the calculation convergence effect is not ideal. Of course, it may not be all the problems of the model itself. Based on the research content of others ' literature and the broad applicability of the k-ε model, the k-ε turbulence closure model was finally selected for numerical simulation.

(7) The authors need to clarify how they examined if the flow was fully developed.

Reply In response to the reviewer's comments, we have clarified our methodology for examining whether the flow was fully developed.

(8) Can authors justify the inclusion of computational mesh in detail? However, the mesh dependency studies are not very convincing. The authors may consider using one of the more sophisticated methods, such as the Grid Convergence Index (GCI), which is commonly applied and recommended by the Fluids Engineering Division of the American Society of Mechanical Engineers as a reliable method for mesh independence analysis. This methodology is described in P. J. Roache, "Perspective: a method for uniform reporting of grid refinement studies," 1994 and applied in much research.

Reply We are very grateful for the valuable learning materials provided by the reviewer. We know that the quality of grid division will significantly impact the accuracy and stability of the calculation results. As far as grid division technology is concerned, we will further strengthen learning in the future. In this study, we use the hexahedral meshing method. When considering the quality of meshing, on the one hand, we are based on some essential criteria for the quality of meshing (such as Skewness, Aspect Ratio, Jacobian Ratio, etc.). On the other hand, we try to compare the accuracy of the calculation under different grid sizes by changing the minimum size of the grid unit and observing the data of the measuring points.

(9) Can authors justify the inclusion of boundary conditions in detail?

Reply Thank you for the insightful questions raised by the review expert. In the initial modeling stage, according to the relevant references and engineering practice, we have developed some basic boundary conditions for the model. In future studies, we will focus on justifying the inclusion of boundary conditions mentioned by reviewers.

(10) Additional details on the reliability and significance of the validation test used to evaluate the differences between experimental and numerical simulation values should be included.

Reply Thank you for your question, the operation law of the materials in the coal pipe at the transfer point of the coal conveying system was numerically simulated. An in-depth analysis of the dust generation mechanism and its causes in the belt coal conveying system was conducted, leading to the optimization of process parameters for the unpowered dust removal system. The research findings have been successfully implemented in the retrofit project of the existing belt coal conveying system at the Huaibei Coal Preparation Plant. This implementation effectively curtails dust generation at its source, significantly reduces dust concentration in the working environment, and ensures employees' occupational health and safety. To further validate the accuracy of the simulation results, a similar experimental simulation platform will be constructed for additional verification, as illustrated in Figure 2.

(11) The critical discussion part has to be enlarged. You need strong discussion in your study. It has to be improved to pin-point the actual novelty(s) in this study. The results are only presented, but a proper discussion is missing.

Reply According to the reviewer’s comment, the critical discussion part has been enlarged.

Please refer to the upload attachment material for detailed modification. Lastly, we extend our sincere gratitude for the exceptional and scholarly revision of our manuscript.

---

## [Decision Letter · Decision Letter 1]

18 Feb 2025

PONE-D-24-47959R1Numerical Simulation Study on Optimization of Key Technical Parameters of Unpowered Dust Removal System in a Gas-solid Two-phase Flow FieldPLOS ONE

Dear Dr. Lv,

Thank you for submitting your manuscript to PLOS ONE. After careful consideration, we feel that it has merit but does not fully meet PLOS ONE’s publication criteria as it currently stands. Therefore, we invite you to submit a revised version of the manuscript that addresses the points raised during the review process.

 The authors are suggested to make changes as highlighted by the reviewer, in particular add details of the selected models, scheme and specified boundary conditions with justification.

We look forward to receiving your revised manuscript.

Kind regards,

Muhammad Shakaib, PhD

Academic Editor

PLOS ONE

Reviewers' comments:

Reviewer's Responses to Questions

**Comments to the Author**

1. If the authors have adequately addressed your comments raised in a previous round of review and you feel that this manuscript is now acceptable for publication, you may indicate that here to bypass the “Comments to the Author” section, enter your conflict of interest statement in the “Confidential to Editor” section, and submit your "Accept" recommendation.

Reviewer #1: (No Response)

2. Is the manuscript technically sound, and do the data support the conclusions?

Reviewer #1: No

3. Has the statistical analysis been performed appropriately and rigorously? 

Reviewer #1: N/A

4. Have the authors made all data underlying the findings in their manuscript fully available?

Reviewer #1: No

5. Is the manuscript presented in an intelligible fashion and written in standard English?

Reviewer #1: Yes

6. Review Comments to the Author

Reviewer #1: The numerical simulation is conducted with the Ansys Fluent software. The specificic version of the software should also be declared.

Most of the references cited in the text are used in the introduction. In Section 4 dedicated to explain the continuous and discrete phase only reference [15] is cited and that reference is published in Chinese in the Journal of China Coal Society.

In the numerial methodology some decisions taken should be justified according to the documentation of the program and other relevant publications in the literature.

In the abstract of the paper is declared that a soft collision model and that the Discrete Element Method is used but in the explanation of the discrete phase in section 4 details are not provided.

Ansys Fluent is a CFD solver package that uses the Finite Volume Method. It has the option to activate the Discrete Phase Model to model multiphase dispersed flows. In the Discrete Phade Model in recent versions the program incorporates the capability to model the interaction between particles using the DEM approach. In my opinion is very important to remark that diferences because there are other in-house and commercial solvers that are purely DEM.

I do not understand some sentences in the introduction "In 1986, American scholar Martin Engineering [5] developed Inertial Flow Technology (IFT) ". Martin Engineerng is a company.

I should explain in more the transition from the real geometry to the computational domain used in the simulations. And in that model I would explain in more detail the boundary conditions used. In Figure 3 it can be observed the presence of a dust curtain and a dust filter but when the numerical methodology is presented nothing is said about how they are modeled.

An unsteady simulation is performed but nothing is said about the temporal scheme and the time step used. All the results are presented without making any mention about the time.

Nothing is said about the treatment of wall-bounded turbulent flows. Are you using standard wall functions or a enhanced wall treatment? Does the mesh used satisfied the requirements of yplus values?

In my opinion the subsection 6.2 should be eliminated from the article. A numerical simulation study should only be validated with results published in the literature

7. PLOS authors have the option to publish the peer review history of their article (what does this mean? ). If published, this will include your full peer review and any attached files.

**Do you want your identity to be public for this peer review?** For information about this choice, including consent withdrawal, please see our Privacy Policy .

Reviewer #1: No

---

## [Author Response · Author response to Decision Letter 2]

4 Apr 2025

Dear Prof. Muhammad Shakaib,

Thank you very much for your letter and the comments from the referees about our paper submitted to Plos One.

We have checked the manuscript and revised it according to the comments. We submit here the revised manuscript as well as a list of changes.

If you have any question about this paper, please don’t hesitate to let me know.

Sincerely yours,

Lv Chen

Response to Reviewer 1:

Thanks for your comments on our paper. We have revised our paper according to your comments:

(1) The numerical simulation is conducted with the Ansys Fluent software. The specificic version of the software should also be declared.

Reply According to the reviewer’s comment, We have provided the specific version of the software.

(2) Most of the references cited in the text are used in the introduction. In Section 4 dedicated to explain the continuous and discrete phase only reference [15] is cited and that reference is published in Chinese in the Journal of China Coal Society.

In the numerical methodology some decisions taken should be justified according to the documentation of the program and other relevant publications in the literature.

Reply According to the reviewer’s comment. In the section 4 of this study, we present a comprehensive review of relevant published literature on gas-solid coupling and other numerical simulation methods.

(3) In the abstract of the paper is declared that a soft collision model and that the Discrete Element Method is used but in the explanation of the discrete phase in section 4 details are not provided.

Reply According to the reviewer’s comment. We restructured the abstract and provided a more comprehensive foundation for the collision calculation in Section 4.

(4) Ansys Fluent is a CFD solver package that uses the Finite Volume Method. It has the option to activate the Discrete Phase Model to model multiphase dispersed flows. In the Discrete Phade Model in recent versions the program incorporates the capability to model the interaction between particles using the DEM approach. In my opinion is very important to remark that diferences because there are other in-house and commercial solvers that are purely DEM.

Reply Thanks the reviewer’s comment. The recommendations provided by the evaluation experts were incorporated into the corresponding sections of the article with appropriate modifications to their content.

(5) I do not understand some sentences in the introduction "In 1986, American scholar Martin Engineering [5] developed Inertial Flow Technology (IFT) ". Martin Engineerng is a company.

Reply Thank you for the insightful questions raised by the review experts. We have reorganized the content of the introduction and thoroughly examined this type of issue.

(6) I should explain in more the transition from the real geometry to the computational domain used in the simulations. And in that model I would explain in more detail the boundary conditions used. In Figure 3 it can be observed the presence of a dust curtain and a dust filter but when the numerical methodology is presented nothing is said about how they are modeled.

Reply Thanks to the valuable questions raised by the review experts, the dust curtain in the model was configured based on the porous medium theory. Specifically, the viscous resistance coefficient was set to 5×e^7 m⁻², the inertial resistance coefficient was set to 80 m⁻¹, and the downstream surface of the dust curtain was designated as a "Trap."

(7) An unsteady simulation is performed but nothing is said about the temporal scheme and the time step used. All the results are presented without making any mention about the time.

Reply Thanks to the valuable questions raised by the review experts, the setting parameter of time step is supplemented in this paper.

(8) Nothing is said about the treatment of wall-bounded turbulent flows. Are you using standard wall functions or a enhanced wall treatment? Does the mesh used satisfied the requirements of yplus values?

Reply Thanks to the valuable questions raised by the review experts, “Non-Equilibrium Wall Functions” are used. The mesh used satisfied the requirements of y+ values, the height of the first layer grid is 0.15 mm, and the grid growth rate is 1.2.

(9) In my opinion the subsection 6.2 should be eliminated from the article. A numerical simulation study should only be validated with results published in the literature

Reply We extend our gratitude to the experts for their valuable suggestions. After careful deliberation, we have decided to retain the content of Section 6.2. The purpose of retaining this section is not to validate the accuracy of the simulation results but to evaluate the actual dust removal performance of the unpowered dust collector under the current research parameter settings.

Lastly, we extend our sincere gratitude for the exceptional and scholarly revision of our manuscript.

---

## [Decision Letter · Decision Letter 2]

24 Apr 2025

Numerical Simulation Study on Optimization of Key Technical Parameters of Unpowered Dust Removal System in a Gas-solid Two-phase Flow Field

PONE-D-24-47959R2

Dear Dr. Lv,

We’re pleased to inform you that your manuscript has been judged scientifically suitable for publication and will be formally accepted for publication once it meets all outstanding technical requirements.

Kind regards,

Muhammad Shakaib, PhD

Academic Editor

PLOS ONE

Additional Editor Comments (optional):

Reviewers' comments:

Reviewer's Responses to Questions

**Comments to the Author**

1. If the authors have adequately addressed your comments raised in a previous round of review and you feel that this manuscript is now acceptable for publication, you may indicate that here to bypass the “Comments to the Author” section, enter your conflict of interest statement in the “Confidential to Editor” section, and submit your "Accept" recommendation.

Reviewer #1: All comments have been addressed

Reviewer #2: All comments have been addressed

2. Is the manuscript technically sound, and do the data support the conclusions?

Reviewer #1: Yes

Reviewer #2: Yes

3. Has the statistical analysis been performed appropriately and rigorously? 

Reviewer #1: N/A

Reviewer #2: Yes

4. Have the authors made all data underlying the findings in their manuscript fully available?

Reviewer #1: (No Response)

Reviewer #2: Yes

5. Is the manuscript presented in an intelligible fashion and written in standard English?

Reviewer #1: Yes

Reviewer #2: Yes

6. Review Comments to the Author

Reviewer #1: Thanks for conducting research in technologies aimed to reduce dust pollution in industry. I am aware that conducting a multiphase CFD simulation involving a complex geometry requires many hours of intense work.

Reviewer #2: I have thoroughly read the revised paper. The authors have applied the modifications proposed by reviewers, and the quality of the manuscript has improved. I would then recommend the publication of this manuscript upon approval by the other reviewers.

7. PLOS authors have the option to publish the peer review history of their article (what does this mean? ). If published, this will include your full peer review and any attached files.

**Do you want your identity to be public for this peer review?** For information about this choice, including consent withdrawal, please see our Privacy Policy .

Reviewer #1: No

Reviewer #2: No

---

## [Editor Report · Acceptance letter]

PONE-D-24-47959R2

PLOS ONE

Dear Dr. Lv,

I'm pleased to inform you that your manuscript has been deemed suitable for publication in PLOS ONE. Congratulations! Your manuscript is now being handed over to our production team.

Kind regards,

on behalf of

Dr. Muhammad Shakaib

Academic Editor

PLOS ONE